# Design constraints for Unruh-DeWitt quantum computers

**Eric W. Aspling[1], John A. Marohn[2,3] and Michael J. Lawler[1,4,5]**

**1** Department of Physics, Applied Physics, and Astronomy,
Binghamton University, Binghamton, NY 13902
**2** Department of Chemistry and Chemical Biology, Cornell University, Ithaca, NY 14853
**3** Department of Materials Science and Engineering, Cornell University, Ithaca, NY 14853
**4** Department of Physics, Cornell University, Ithaca, NY 14853
**5** Department of Physics, Harvard University, Cambridge, MA 02138

## Abstract

The Unruh-DeWitt particle detector model has found success in demonstrating quantum information channels with non-zero channel capacity between qubits and quantum fields. These detector models provide the necessary framework for experimentally realizable Unruh-DeWitt quantum computers with near-perfect channel capacity. We propose spin-qubits with gate-controlled coupling to Luttinger liquids as a laboratory setting for Unruh-DeWitt detectors and explore general design constraints that underpin their feasibility in this and other settings. We present several experimental scenarios including graphene ribbons, edges states in the quantum spin Hall phase of HgTe quantum wells, and the recently discovered quantum anomalous Hall phase in transition metal dichalcogenides. Theoretically, through bosonization, we show that Unruh-DeWitt detectors can carry out quantum computations and identify when they can make perfect quantum communication channels between qubits *via* the Luttinger liquid. Our results point the way toward an all-to-all connected solid state quantum computer and the experimental study of quantum information in quantum fields *via* condensed matter physics.

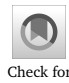

# 1  Introduction

Unruh-DeWitt (UDW) detectors originated as a thought experiment by Unruh [1] (later extended by DeWitt [2]) to model an accelerating qubit in a vacuum. Unruh showed that an accelerating observer would view the ground state of a quantum field as a mixed state and see a loss of quantum information hidden behind the Killing horizon [3]. Today, there are parallel efforts to utilize UDW detectors to advance science in cosmology, high energy physics, and condensed matter physics. Cosmologists use them to model information in highly accelerated frames of reference (e.g. in and around black holes), high energy theorists use them without acceleration to study quantum information flow *via* quantum fields, a field called relativistic quantum information [4–10]. Condensed matter experimentalists use UDW detectors such as nitrogen-vacancy centers and superconducting interference devices, calling the detectors "quantum sensors", to sensitively detect electromagnetic fields produced by a wide variety of systems from quantum materials to systems outside of condensed matter like cancer cells. But currently, experimentalists demand much less from the UDW detectors than theorists, having yet to use them to study the flow of quantum information in complex systems.

A feature that theorists require of UDW detectors is the ability to turn their coupling to their environment on and off rapidly. Consider a spin qubit coupled to a one-dimensional wire modeled as a Kondo-like impurity. A simple look at such a quantum computer is displayed in Fig. 1 which showcases natural scalability as a feature that preserves all-to-all quantum communication. A setup such as this is capable of sensing the flow of small currents in the wire. Turning the Kondo-like coupling on and off rapidly, however, turns it into an emitter that sends signals through the wire, signals to be picked up by another such spin qubit acting as a detector. The net result of this communication amounts to a quantum gate that acts unitarily on the combined qubit-wire system. In Fig. 7, we take this idea to the next level: the design of an all-to-all connected solid-state quantum computer where gates can be applied to distant qubits enabled by communication *via* quantum coherent wires. The possibility that a quantum wire could achieve such communication dates back to at least as early as 2007 [11,12]. Control over timing, therefore, enables the UDW detector to emit and receive quantum information. This propagation of quantum information offers clear benefits to quantum technology.

In addition to practical applications in quantum computing and communication, timing-controllable UDW detectors would allow the study of complex quantum systems in a new way. Careful construction of quantum information channels through these systems allows for

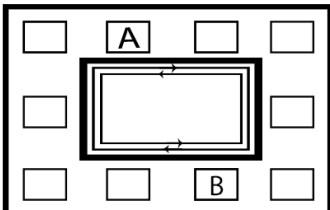

(a) Qubit A can access all qubits on the system *via* left- and right-movers.

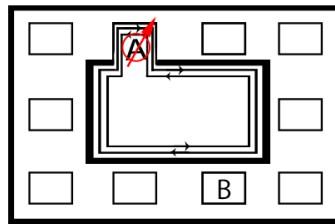

(b) Opening up qubits will cause interactions with the left- and right-movers.

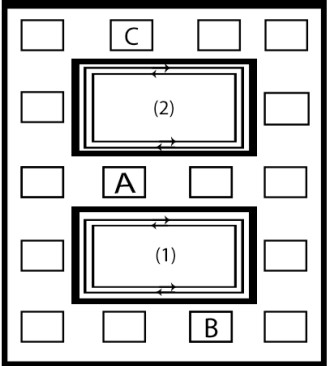

(c) Adding another block of qubits restricts of our left- and right-movers to paths (1) or (2).

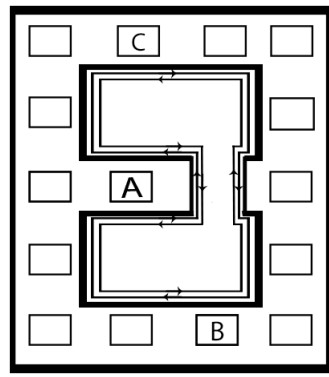

(d) Opening the bulk between the blocks gives Qubits B and C direct access to the entirety of the qubits.

Figure 1: Figures (a-d) show a simplistic view of our quantum bus. Qubits (such as qubits A and B) are placed around the edges of a Luttinger liquid. The potential wall is lowered, as in Fig. (b) and the qubits are able to interact with the topological edge states. Figures (c) and (d) indicate how scaling up this system can be done easily by adjusting where the bulk of our fields lives through raising and lowering the potential barrier.

a deeper understanding of their quantum properties without directly carrying out projective measurements on the system or inferring them through measurements of expectation values. Simulating channel capacity, for example, would show how entanglement spreads within them and how quantum information becomes scrambled. If we could similarly study physical systems, we could directly probe their entanglement properties.

Studying quantum information channels into and out of complex quantum systems provided by UDW detectors will also enable these systems to become part of quantum technology. For example, a system described by a quantum field could become a component in a quantum computer that can carry out computations (these fields are known as flying qubits). The grand application of such a computer would then be to simulate quantum field theory, a task long predicted to be a consequence of quantum computing [13–16]. Such a simulator, for example, could simulate Dirac fermions directly without needing to overcome the fermion doubling problem caused by discretization.

One possible system to achieve simulation of Dirac fermions is known as the Yao-Kivelson model, which has been studied in Ref. 17. The Yao-Kivelson model provides a solvable model of topological edge states that can be expressed as Majorana fermions. Dangling qubits near the edge state provide a comparable model to a UDW detector and offer helpful insights regarding

restrictions to our system. In Sec. 2.2 we discuss this system as it pertains to our own in more detail. In the near term, we expect a UDW quantum computer that utilizes a quantum bus like that of Fig. 1 to be better equipped at enhancing error-correcting codes by exploiting all-to-all connectivity. So we see that timing-controlled UDW detectors will have both fundamental and technological applications.

In this paper we propose and assess three potential systems for realizing the study of quantum information flow *via* timing-controlled UDW detectors coupled to quantum materials: HgTe/CdTe heterostructures in the quantum spin Hall phase [18, 19], graphene spin qubits [20–24], and Moiré transition metal dichalcogenides (TMDs), in either the anomalous or spin quantum-Hall regime [25, 26]. Each of these systems contains spin qubits coupled to Luttinger liquids. We begin with a theoretical analysis combining the formalism of UDW detectors with the bosonization of Luttinger liquids. This innovation gives us the ability to engineer new systems for exploring quantum information flow through quantum fields. Using the results of Simidzija *et al*. [5–7], we show that a non-zero channel capacity should exist in these systems. We provide a library of Hamiltonians that characterize the qubit-field quantum transduction constraints demonstrated in this paper, bolstering the natural viability of quantum electronics/communication. We assess the experimental viability of the three proposed systems. We close by discussing future theoretical work, outlining the many new avenues of information research generated by the timing-controlled UDW detectors proposed here.

## 2 Quantum information flow in quantum materials

The trend for scaling up quantum computing consists of larger and larger numbers of qubits carrying out linear operations over longer length scales. This approach at scaling seems natural as error correcting codes are more accurate with more qubits [27–29]. However, topological systems such as the fractional quantum Hall effect could provide a quantum bus of all-to-all connected qubits which offer a robust error correcting code [13] that scales at least like Fig. 1. The all-to-all communication channels provide situational error-correcting codes based on stabilizer codes [30] and the periodic condition of our quantum bus resolve length scales. The most important question is thus: how well does our system process quantum information? Undertaking this task involves devising quantum channels displaying maximal channel capacity.

### 2.1 Devising the quantum channel

Quantum channels present the necessary formalism to analyze entanglement propagation through a quantum circuit [31, 32]. Recently, high-energy physicists have made progress investigating quantum channels as they exist between fields and qubits. As mentioned in the introduction, they use UDW detector formalism that utilizes a smearing function to spread a two-dimensional Hilbert space onto an infinite-dimensional space. This formalism carries a series of complications such as limitations due to the no-cloning theorem and information spreading due to Huygen's principle in spatial dimensions higher than one [5, 33]. In this regard, an analogy to classical wireless communication is not possible. Instead, advancing quantum devices such as quantum wires may prove more profitable.

Using quantum gate formalism, we evaluate the quantum coherent information, a figure of merit for the channel, which is analogous to mutual information in classical information theory. For the systems discussed in this paper, quantum coherent information is computed according to the circuit shown in 2. The construction of candidate unitaries, such as $U_A$ and $U_B$ in Fig. 2, needed for our quantum channel, is done carefully in Sec. 3. For now, we will

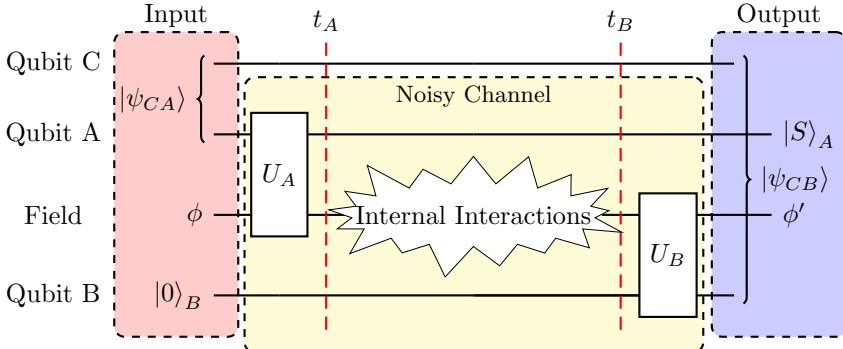

Figure 2: Quantum coherent information is a measure of quantum information flowing through a quantum channel. To compute it, we use the above circuit diagram. Initially, Qubit A is entangled with Qubit C in a Bell state. Then Qubit A coupled to the Field $\phi$ through $U_A$ and later Qubit B $|0\rangle_B$ is similarly coupled through $U_B$. Finally, the entanglement is measured between Qubit B and Qubit C. If these qubits are entangled, then the coherent information is positive and quantum information flowed through the noisy channel.

outline some design constraints of these unitaries that enable them to successfully transport quantum information.

Simidzija *et al.* as well as others, have recently laid the groundwork for models that successfully outline the necessary conditions of just such a channel [5–7,34]. They have shown that UDW unitaries which behave as controlled unitary gates, lead to entanglement-breaking channels when processed alone. However, carefully applying two of these rank-one unitaries breaks the controlled-gate structure of the circuit, allowing them to properly encode (or decode) information from a spin structure onto a scalar bosonic field. In other words, the channel created by gates with these unitaries can have a positive-valued coherent information that scales with coupling and smearing parameters.

More elusive is a schematic for strongly coupled fermionic systems using UDW detectors. As we will demonstrate, the formalism describing quantum channels, traces over the field and results with a correlator of field operators. Mapping fermions to bosons through bosonization has an equivalence at the level of the correlators. We find that through the bosonization of our Luttinger liquid, we can create different approaches to solve this problem. Section 3 aims to provide a library of these gates which enable channels with non-zero capacity. Furthermore, we claim that careful parameter selection can theoretically create a near-perfect quantum channel.

## 2.2 Design parameters governing channel performance

There are many parameters governing a field-mediated quantum channel between two qubits. We can separate the channel into two components, gates between the qubit and field and the propagation pathway the quantum information travels along within the field itself.

Generally, a gate between a qubit and a field is governed by a coupling function $J(x,t)$. We can break this function into three factors as is common in the relativistic quantum information literature. One is a switching function $\chi(t)$ normalized to $\int dt\, \chi(t) = 1$ that turns the gate on and off. Another is a smearing function $p(x,y)$ that couples a qubit at $x$ to the field at various points $y$. It too is normalized with $\int dy\, p(x,y) = 1$. Ideally, both $\chi(t)$ and $p(x,y)$ are non-negative functions. Presumably, $p(x,y)$ is non-zero only inside the quantum dot (Qdot) hosting the qubit. Finally, there is the overall dimensionless strength of the coupling $J$. Hence the coupling function $J(x,t)$ is naturally parameterized by this strength $J$, a switching time $t_{\text{sw}}$ and a smearing length $\lambda_s$.

For our models in Sec. 3, we use a Dirac delta-like switching function with $t_{sw} = 0$, a mathematically convenient but physically impossible situation. It allows the gate to produce a change in the field that remains perfectly localized within the smearing length. For $t_{sw} > 0$, during the action of the gate information will spread away from the qubit at the velocity $v$, the effective speed of light governing the relativistic field. Hence, if we have $t_{sw} < \lambda_s/v$, the effect of the gate will remain nearly localized within the smearing length, and a physically realizable gate will behave similarly to our idealized gates.

Given $\lambda_s$, $t_{sw}$, and $v$, we now have two dimensionless parameters governing the design of a gate: the coupling strength $J$ and the *gate-localization quality* $Q_{loc} = t_{sw}v/\lambda_s$. A small $Q_{loc}$ is a *design constraint* for a UDW quantum computer. If it is too large, quantum information will spread over large distances and during the action of the gate making, it is difficult to recapture. A small $J$, however, implies the gate has little effect. Hence for good channel performance, we will want gates with a small $Q_{loc}$ and large $J$.

Earlier studies, Refs. 17, 35 identify another design constraint of quantum information channels in condensed matter physics. Though different from an UDW Quantum Computer, Refs. 17, 35 provide an approach to carry out such a process by modeling a dangling qubit near a topologically protected edge state or an end spin of a spin chain. Their perturbative approach offers no analytic limits of a perfect quantum information channel (a proof can be found in the discussion surrounding Eq. 16 of Ref. 5). However, it offers other insights, including *identifying internal interactions that promote scrambling*. This study therefore highlights an important design criteria: to study quantum information in a quantum material, it must flow and be picked up within the scrambling time $t_s$ of the material or it will be lost.

We therefore need to understand how, as the information propagates, it is subject to scrambling by interaction effects [36–38]. Measurements on the target qubit may detect the onset of quantum chaos caused by the system's inherent disorder [36,39,40]. Similarly, if the information runs into a magnetic impurity acting like an uncontrolled qubit, it may be stolen by it and never reach the intended qubit [41]. The information could also be taken away by phonons and spread throughout the host material [42]. So this intermediate stage is simultaneously an opportunity to study the physics of the host quantum material at a quantum information level and a constraint on the performance of the communication —*it limits the distance between communicating qubits to $v/t_s$*.

Given the above design constraints, we next turn to the question; in ideal circumstances, does a perfect communication channel exist for qubits coupled to Luttinger liquids?

# 3 Coupling quantum dots and Luttinger liquids to create Unruh-DeWitt detectors

## 3.1 Dirac fermions meet qubits

### 3.1.1 Reframing the Unruh–DeWitt Hamiltonian

Massless Dirac fermions evaluated in UDW detector models have been a promising endeavor for relativistic quantum information processes [34,43,44]. However, scalar field theories have been more successful in producing simulations of quantum information channels in relativistic quantum information [5,7,34]. In this section, we aim to show that bosonization is a tool that provides a convenient bridge between these two approaches.

Unruh–DeWitt detectors are commonly used when coupling a two-level system to a field [9]. This can be utilized in one of two ways. Firstly, an observer (detector) experiencing acceleration is subjected to radiation (the Unruh effect [3]) that an inertial observer would not

be exposed to. The detector would then be in an excited state as it interacts with the thermal bath of radiation. The second usage of UDW detectors for qubit-field couplings exists as an interaction that passes excitations back and forth between qubits and fields. This excitation can include quantum information in a method known as "entanglement harvesting" [6]. For this note, we specifically utilize the second scenario, with intent to design field mediated communication channels through LLs.

In non-perturbative theories, they are most readily studied if the coupling is linear. In the case of a helical Luttinger liquid (HLL), the field is a 1+1 dimensional Dirac fermion, but a linear coupling to a Dirac fermion would violate fermion number conservation [34]. Fortunately, when quadratically coupled to this field, the interaction can be modeled as linearly coupled to a bosonic field, *through bosonization*. Hence, bosonization is a powerful method to aid the analysis of UDWs in Luttinger liquids. Specifically, we relate fermions and bosons at the level of the correlator as this is the result of tracing over the field in the formalism of quantum channels [5,31]. At the level of the correlator, bosonization is a true duality between fermions and bosons [45].

To describe UDWs in HLLs, we need to define a physical two-level system and find its coupling to the HLL. The redundancy in chiral indices of a HLL model gives us a "spinless-like" model [46,47], but spin is still physically present. Spin-up travels one way around the edge of the system, our "+" mover, while spin-down travels the other, our "−" mover. So the simplest two-level system would be a spin qubit with a finite spatial extent $p$ that we take to be approximately Gaussian

$$p(x, y) = \frac{1}{\sqrt{2\pi\sigma^2}} e^{-\left(\frac{(y-x)^2}{2\sigma^2}\right)}. \tag{1}$$

If the spin qubit is a single atom, it would presumably have a $\sigma$ of order the size of the atom, lattice spacing, and $k_F^{-1}$. But if synthesized as a quantum dot, $\sigma$ would be the size of the dot, and thus much larger in extent than the atomic scale.

Introducing a Dirac delta type switching function $\chi(t)$ that provides control over the coupling between the qubit and field, yields a UDW detector Hamiltonian with a z-component Kondo-like interaction

$$H_{\text{int}}(t) = \chi(t) \int_{\mathbb{R}} dy \, p(x(t), y) J_{\alpha,z} \mu_{\alpha}(t) (\psi_+^\dagger(y)\psi_+(y) - \psi_-^\dagger(y)\psi_-(y)). \tag{2}$$

Here $\psi_-$ ($\psi_+$) denote our spin-down, left-moving (spin-up, right-moving) fermions. This interaction term assumes the Qdot limit where factors of $e^{\pm 2ik_F y}$ have been suppressed as we restrict $\sigma >> \frac{1}{k_F}$ [48]. We will return to this assumption in Sec. 3.3. Notice that Eq. 2 has many familiar properties, but the introduction of the smearing and switching functions demonstrate the exact form of a UDW Hamiltonian [49]. Since this model is a Kondo-like model, some may recognize our two-level system as the Kondo-like impurity given by

$$J_{\alpha,z}\mu_{\alpha}(t) = J_{xz}S_x(t) + J_{yz}S_y(t) + J_{zz}S_z(t) \tag{3}$$

$$= \frac{J}{2}(S_+ e^{-i\Omega t} + S_- e^{+i\Omega t}) \equiv J\mu(t). \tag{4}$$

Where the second equality follows by choosing $J_{\alpha,z} = J\hat{X}_\alpha$ to point along a new $\hat{X}$ direction and time dependence generated by Hamiltonian $H_0 = -g\mu_B \vec{B} \cdot \vec{S} \equiv \Omega S_Z$ with $\hat{Z}$ perpendicular to $\hat{X}$. We've added the magnetic field to show that this system is a two-level system.

An important point to notice here is that the coupling presented above is not exact to that of a Kondo impurity. We have specifically ignored physical effects such as RG Flow and Kondo screening. Instead we utilize the dimensionless coupling constant above as having no quantum corrections for simplicity. However, stress, strain, and temperature have potential to

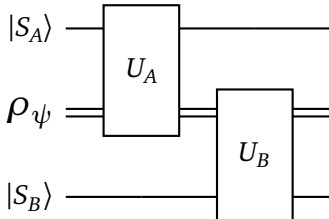

Figure 3: $|S_i\rangle$ is the state of the Qdot and $\rho_\psi$ represents the state of the left and right chiral fermions.

dramatically alter the effective coupling utilized. To realize the experimental systems proposed in Sec. 4, these theoretical questions need to be further investigated.

Our formulation takes the usual convention of a HLL and implants it into the UDW model to describe a quantum channel between distant spin qubits communicating *via* Dirac fermions. It allows for an elegant promotion to a quantum circuit model by constructing the Hamiltonian in the following way

$$H_{\text{int}}^{(r,\nu)}(t) = J\chi(t)\mu(t) \otimes O_{\nu r}^{\dagger}(t)O_{\nu r}(t), \tag{5}$$

where $\nu$ specifies the interacting spin-qubit ($\nu \in \{A,B\}$ in Fig. 1), $r \in \{\pm\}$, and we absorb the smearing functions into the field observables $O_{\nu r}$ as is common in detector literature. The delta-like switching function $\chi(t)$, can be seen as the instance of time $t_\nu$ of interaction for qubit $\nu$ such that we can express the coupling as $J_\nu = J\chi(t_\nu)$. We can then construct a set of unitary operators $U_A$ and $U_B$ that take the form

$$U^{(\nu)} = \exp\left(-iJ_\nu\mu_\nu(t) \otimes O_{\nu r}^{\dagger}(t)O_{\nu r}(t)\right). \tag{6}$$

### 3.1.2 Unruh–DeWitt quantum channels

It is crucial that a channel utilizing this gate (depicted in the circuit diagram given by Fig. 3) can propagate the entanglement. If we encode information onto quantum fields through gate $U_A$ at time $t_A$ we need to decode that information effectively through $U_B$ at some later time $t_B$. To determine the effectiveness of this channel we can utilize the metric of quantum capacity. Following the prescription of Sec. IV in Ref. [5] we find that the quantum channel and subsequently quantum capacities rely on field correlators. Bosonization allows one to replace fermionic operators with a bosonic counterpart, which are equivalent at the level of the correlator [45].

## 3.2 Bosonizing the Hamiltonian

As mentioned in Sec. 1, our goal here is to evaluate if a quantum channel through our system has a non-zero channel capacity. One method to explore channel capacity is to find if the states are separable at any given point in the circuit. If separable states exist, then we are left with classical information. The channel is then said to be a "quantum-to-classical" channel (i.e. the channel is entanglement breaking [31]). As mentioned previously, the literature points us in a direction where non-zero capacity quantum channels can be realized through unitary gates. However, the fields employed for strongly coupled processes are bosonic.

Working in the interaction picture, as we are above, is enabled in bosonization by multiplying the fermionic creation and annihilation operators by a phase of $e^{\pm i\nu|k|t}$, so long as our left- and right-moving bosons in Eq. 8 are massless [50]. Below, we introduce a quick primer to the bosonization process, while mostly remaining in the interaction picture.

When bosonizing our fermionic fields, we utilize the interaction picture and consider variable $z$ with $z = -i(x - vt)$ and $\bar{z} = i(x + vt)$. Without the interaction picture, the point-splitting process used to evaluate the bosonized forms of our fermionic fields will artificially eliminate necessary terms. At the level of the unitary gates, we can consider the Schrödinger picture, as our switching function is of delta form.

Standard bosonization procedures [45, 47, 50, 51] to find bosonized fermionic operators are of the form

$$\psi_+ \to \frac{1}{\sqrt{2\pi}} e^{-i\sqrt{4\pi}\phi(z)}, \qquad \psi_- \to \frac{1}{\sqrt{2\pi}} e^{i\sqrt{4\pi}\bar{\phi}(z)}. \tag{7}$$

Where $\phi(z)$ is a real scalar field given in the interaction picture by

$$\phi(z) = \int_{k>0} \frac{dk}{2\pi} \frac{1}{\sqrt{2k}} [b(k)e^{-kz} + b^\dagger(k)e^{kz}], \tag{8}$$

$$\bar{\phi}(\bar{z}) = \int_{k>0} \frac{dk}{2\pi} \frac{1}{\sqrt{2k}} [\bar{b}(k)e^{-k\bar{z}} + \bar{b}^\dagger(k)e^{k\bar{z}}]. \tag{9}$$

Consistent with the literature, these fields are a natural result of the mode expansions

$$\varphi(x) = \int \frac{dk}{2\pi} \sqrt{\frac{v}{2\omega(k)}} [b(k)e^{ikx} + b^\dagger(k)e^{-ikx}], \tag{10}$$

$$\Pi(x) = \int \frac{dk}{2\pi} \sqrt{\frac{\omega(k)}{2v}} [b(k)e^{ikx} + b^\dagger(k)e^{-ikx}], \tag{11}$$

with $\omega(k) \equiv v|k|$. Equation 10 is related to $\phi$ through a duel boson $\vartheta$ by $\phi = \frac{1}{2}(\varphi + \vartheta)$ and $\bar{\phi} = \frac{1}{2}(\varphi - \vartheta)$. Under this formalism, the normal-ordered density operators become $\rho_- =: \psi_-^\dagger \psi_- := -\frac{i}{\sqrt{\pi}}\partial_z \phi$ which allows us to rewrite our Hamiltonian linearly as

$$H_{\text{int}}^{\text{Bos}}(t) = J_v \int_{\mathbb{R}} dy \, p(x(t), y)\mu(t)\left(\frac{1}{\sqrt{\pi}}(\partial_z \phi + \partial_{\bar{z}} \bar{\phi})\right) \tag{12}$$

$$= J_v \int_{\mathbb{R}} dy \, p(x(t), y)\mu(t)\left(\frac{1}{\sqrt{\pi}}\Pi\right). \tag{13}$$

We can see here that the right- and left-movers can be combined into a single equation that provides us with a simple rank-one unitary gate. After including the smearing function into our conjugate momentum field $\Pi$, equation 6 becomes

$$U^{(v)} = \exp(iJ_v \mu_v \otimes \Pi_v). \tag{14}$$

This simple rank-one unitary is nice for transferring information onto and off of the field, but to build a quantum channel that does not break entanglement we need more.

## 3.3 A library of gates

### 3.3.1 Simple rank one unitary gates

We have in Eq. 14 the first gate of our quantum computer. If we consider a channel that connects Qubit A directly to Qubit B (as described in Fig. 1(a)), then in essence we have created a channel, which at some point processes classical information. A "measurement" takes place [31]. In order to effectively transfer entanglement onto and off of our fields we need to have, minimally, two rank-one simple unitary transformations [5, 7],

$$U^{(v)} = \exp(iJ_{v2}\mu_{v2} \otimes O_{v2})\exp(iJ_{v1}\mu_{v1} \otimes O_{v1}). \tag{15}$$

The first naive attempt at expressing our Hamiltonian as two rank-one unitaries may be understood better if we ignore the time-reversal symmetry of our system and instead write a new combination of field densities. More specifically, we set up a toy model to designed to realize this task. Let us revisit the relationships we discuss in Sec. 3.2. Namely,

$$\rho_+ =: \psi_+^\dagger \psi_+ : = \frac{i}{\sqrt{\pi}}\partial_z \phi \,,$$

$$\rho_- =: \psi_-^\dagger \psi_- : = -\frac{i}{\sqrt{\pi}}\partial_{\bar{z}} \bar{\phi} \,,$$

(16)

which provides us with two field densities that are bosonized in the following way:

$$\rho = \rho_+ + \rho_- = \frac{1}{\sqrt{\pi}}\partial_x \varphi \,,$$

$$j = \rho_+ - \rho_- = \frac{1}{\sqrt{\pi}}\Pi \,.$$

(17)

Notice now, if we choose our coupling carefully, we can craft an interaction Hamiltonian of which the bosonized form is

$$H_{\text{int}}^{\text{Naive}}(t) = \int_{\mathbb{R}} dy\, p(x(t),y)(J_+\mu_+(t)\rho + J_-\mu_-(t)j)$$

(18)

$$= \int_{\mathbb{R}} dy\, p(x(t),y)\frac{1}{\sqrt{\pi}}(J_+\mu_+(t)\partial_x\varphi + J_-\mu_-(t)\Pi),$$

(19)

and yields a gate similar to 15 in the form of

$$U^{(\nu)} = \exp(iJ_-\mu_- \otimes \Pi)\exp(iJ_+\mu_+ \otimes \partial_x\varphi).$$

(20)

### 3.3.2 Chiral Luttinger liquid gates

For the naive Hamiltonian, we were explicit about breaking our time-reversal symmetry. The remainder of the gates in our library we focus on systems that more likely to be physically realizable but leave the numerical quantum capacity simulations for future work.

Consider for example, our Hamiltonian from Eq. 2. If we were to rewrite this equation with separated left- and right-mover channels as

$$H_{\text{int}}^{\text{LR}}(t) = \int_{\mathbb{R}} dy\, p(x(t),y)(J_\alpha\mu_\alpha(t)\psi_+^\dagger\psi_+ - J_\beta\mu_\beta(t)\psi_-^\dagger\psi_-),$$

(21)

where $J_\alpha = 0$ for left-movers only, and $J_\beta = 0$ to suppress right-movers then we find a similar construction to Eq. 12 without the ability to combine the fields as we did previously. From here we can construct conjugate momentum $\frac{1}{\nu}\partial_t\phi$ and $\frac{1}{\nu}\partial_t\bar{\phi}$ for right- and left-movers respectively. Then evaluate an "all left-moving channel" or an "all right-moving channel", yielding the same form as Eq. 20. Experimentally this construction may be accomplished by restricting how the Qdot interacts with the HLL or through a chiral Luttinger liquid such as found in the recently discovered anomalous quantum Hall effect in Moiré heterostructures [25].

### 3.3.3 Including the cross-terms

Another gate might be found in the cross terms suppressed by the factor of $e^{\pm 2iK_F x}$. If instead of suppressing these interactions we allow the backscattering terms á la Ref. [48], we gain

coupled degrees of freedom that yield promising unitaries as well. These terms written out explicitly take the form

$$H_{\text{int}}^{\text{CT}}(t) = J_\nu \int_{\mathbb{R}} dy \, p(x(t), y)\mu(t)(e^{-i2K_F x}\psi_+^\dagger \psi_- - e^{+i2K_F x}\psi_-^\dagger \psi_+). \tag{22}$$

Using the above definitions of our bosonized fermions with added Klein factors to preserve anticommutation relations of the fermions [45, 50, 51], we find a bosonized Hamiltonian

$$H_{\text{int}}^{\text{CT}} = J_\nu \mu(t)\left(\frac{1}{2\pi}\cos\sqrt{4\pi}(\varphi)\right). \tag{23}$$

Combining Eqs. 23 and 13 provides a very interesting but non-linear interaction. A unitary gate

### 3.3.4 Non-chiral Luttinger liquid gates

Often when considering the gates in Sec. 3.3.3, one wants to include both the spin and charge sectors. This scenario describes Dirac fermions that are free to spin and propagate in either direction. Following the usual bosonization prescription, we introduce two bosons $\varphi_\uparrow$ and $\varphi_\downarrow$, that are related by the charge and spin bosons

$$\varphi_c = \frac{1}{\sqrt{2}}(\varphi_\uparrow + \varphi_\downarrow), \qquad \varphi_s = \frac{1}{\sqrt{2}}(\varphi_\uparrow - \varphi_\downarrow), \tag{24}$$

as well as chiral fields $\phi_{c,s}$ and $\bar{\phi}_{c,s}$. Using these definitions, we can see that our bosonized Hamiltonian can be split into two sections, forward-scattering and back-scattering. The forward-scattering terms are without the factors of $e^{\pm 2ik_F x}$, and simplify using the same point-scattering methods used in deriving Eq. 13. The forward-scattering bosonized Hamiltonian is given by

$$H_{\text{int}}^{\text{F}} = J_\nu \mu(t)\left(\frac{2}{\sqrt{\pi}}(\Pi_c + \partial_x \varphi_s)\right), \tag{25}$$

which is of the same form as Eq. 19 but our starting point was 2, so this Hamiltonian preserves time-reversal symmetry.

Since the fermion fields in the back-scattering (cross-terms) anticommute, we can straightforwardly bosonize the fermions. The resulting back-scattering Hamiltonian is

$$H_{\text{int}}^{\text{BS}} = J_\nu \mu(t)\left(\frac{1}{2\pi}\cos\sqrt{2\pi}(\varphi_c + \vartheta_s)\right). \tag{26}$$

Notice here if we suppress the spin terms (make the system spinless) we retrieve the same Hamiltonian we would by combining Eqs. 13 and 23. When we consider both spin and charge, we have two noncommuting observables that could be used to create an arrangement of operators to explore novel quantum channels of information.

### 3.3.5 Dirac Hamiltonian gates

Another coupling that may be experimentally present is similar to the Kondo-like coupling of Eq. 2, but instead, a free Dirac fermion according to the Dirac Hamiltonian is coupled to our spin-qubit as follows

$$H_{\text{int}}^{\text{D}}(t) = J_\nu \int_{\mathbb{R}} dy \, p(x(t), y)\mu(t)(\psi_+^\dagger \partial_x \psi_+ - \psi_-^\dagger \partial_x \psi_-). \tag{27}$$

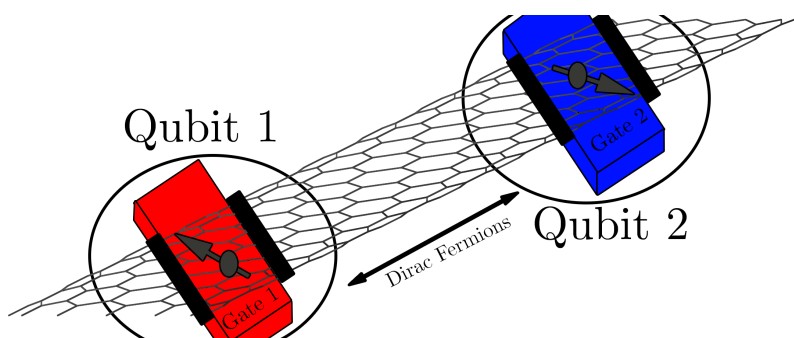

Figure 4: Graphene nanoribbons, as proposed by Ref. 11, as a possible scenario for realizing UDW detectors.

This equation has the well-known bosonized form of [47,50]

$$H_{\text{int}}^{\text{D}} = J_\nu \int_{\mathbb{R}} dy \; p(x(t), y)\mu(t)(\Pi^2 + (\partial_x \varphi)^2). \tag{28}$$

Hamiltonians like Eq. 28 have been addressed with great success through perturbative approaches [34]. Our system, in contrast, provides a unique opportunity to explore the strong coupling of a quadratic interaction. However, since $[\varphi^2(x), \Pi^2(x')] = 2i\{\varphi(x), \Pi(x')\}\delta(x - x')$, exponentiation into simple unitary gates is not as straightforward.

### 3.4 Constructing a realistic quantum channel

We have shown several scenarios where the bosonization of a Luttinger liquid leads to unitary gates like that of Eq. 15. Particularly some of these gates that consist of two rank-one unitaries are of the same form used by Simidzija *et al.* to simulate coherent information, as shown in Fig. 4 of Ref. 5. This result demonstrates that coherent information asymptotically approaches one as the strength of the coupling $J_{\nu 1}$ grows with respect to the width of the Gaussian smearing function $\sigma$.

Furthermore, while our latter systems do not produce the same unitary gates as that of Ref. 5, they are not inherently entanglement-breaking. Future studies of nonlinear Hamiltonians, like those of Eqs. 23 and 28, will be needed to understand if the same relationship is present. Regardless, our toy-model demonstrates a possibility that an electron gas of some 2D materials has the potential to transport quantum information in a near-perfect way up to some corrections.

## 4 Experimental scenarios

There are many scenarios for experimentally realizing a UDW quantum computer. Here we consider three to give a sense of how they might be designed. The first is to upgrade the graphene ribbon proposal of Ref. 11 to define gates between the spin qubits and the conducting channels. The second scenario is to build solid state quantum dots (Qdots) and embed them in a HgTe/CdTe quantum well. The third scenario is to build transition metal dichalcogenide (TMD) qubits and place them in a heterostructure exhibiting the recently discovered quantum anomalous Hall effect phase. Other possibilities include quantum spin chains acting as the field [35, 52] (they can be viewed as fermions through the Jordan-Wigner transformation), and silicon nanowires [53].

Table 1: Temperature $T$, magnetic field $B_0$, electron-spin resonance frequency $f_e = \gamma_e B_0/2\pi$, and thermal spin polarization $p_{th} = \tanh(\hbar\gamma_e B_0/2k_b T)$. In the previous formulas, $\gamma_e$ is the electron gyromagnetic ratio, $\hbar$ is the reduced Planck's constant, and $k_b$ is Boltzmann's constant. In computing $f_e$ and $p_{th}$ we assume $\gamma_e = 2\pi \times 28.0\,\mathrm{GHz\,T^{-1}}$, appropriate for a $g = 2$ electron, and use $\hbar\gamma_e/2k_b = 0.672\,\mathrm{K\,T^{-1}}$.

| $T\,[\mathrm{K}]$ | $B_0\,[\mathrm{T}]$ | $f_e\,[\mathrm{GHz}]$ | $p_{th}$ |
|---|---|---|---|
| 4.2 | 1.4 | 39.2 | 0.22 |
| 4.2 | 4.5 | 126. | 0.62 |
| 4.2 | 9.0 | 252. | 0.89 |
| 2.1 | 9.0 | 252. | 0.99 |

Let us review the graphene ribbon scenario proposed in Ref. 11 and assess its potential for building UDW detectors. Figure 4 presents the scenario: a graphene nanoribbon with gates applied to trap electrons, leaving conducting channels between the gates. The scenario works because of the Klein effect that enhances the conduction of Dirac fermions in the channels instead of suppressing it. The speed of electrons in graphene [54] is between $v = 0.8 \times 10^6\,\mathrm{m\,s^{-1}}$ and $3 \times 10^6\,\mathrm{m\,s^{-1}}$. Using units on the nanoscale, this translates to a slowest velocity of $v = 1 \times 10^6\,\mathrm{nm\,ns^{-1}}$. The qubit, according to Ref. 11, is about 30 nm in length. This qubit size suggests we take the smearing length to be $\lambda_s = 30\,\mathrm{nm}$. Hence, for a gate-localization quality $Q_{loc} \approx 1$ we require a switching time of roughly $t_{sw} = \lambda_s/v = 3 \times 10^{-5}\,\mathrm{ns} = 30\,\mathrm{fs}$.

There is precedent for achieving electrical switching on such a fast timescale using sub-100 fs light pulses, and we can build on this precedent to design our quantum computer. One approach would be to employ a small-area metal electrode as an electrical gate to inject an electron into the qubit, with the gate fabricated on a thin tunneling (i.e. oxide) gap beneath a semiconductor dot. Using the non-linear electrical conductivity of the tunneling gap, electrons could be rapidly injected into the gate using terahertz pulses, as in junction-mixing scanning-tunneling microscope experiments [55–59]. Another approach would be to inject electrons into the gate using optical pulses illuminating a nearby photoswitch [59–62]. While it would be extremely challenging to build electrical waveguides good enough to achieve sub-100 fs gate switching using voltage pulses alone, we note that recent progress in fabricating nanodiodes [63] may make such ultrafast electrical switching feasible in the near future.

In addition to fast switching times, we also need to consider initialization. For initialization, one would need the electron spin of the qubit to be fully polarized, which in turn requires working at high field and low temperature. Table 1 shows the thermal spin polarization $p_{th}$ expected at various temperatures and fields. Also shown is the associated electron spin resonance frequency for a $g = 2$, donor-bound electron. Employing cryogenic chip-scale microwave sources [64,65] operating at cryogenic temperatures [66] would allow one to work at magnetic fields up to $B_0 = 9\,\mathrm{T}$ and therefore at a relatively high temperature of $T = 4.2\,\mathrm{K}$ (liquid helium) or 2.1 K (pumped liquid helium). At $B_0 = 9\,\mathrm{T}$ and $T = 2.1\,\mathrm{K}$ the electron-spin polarization is $p_{th} = 0.99$, and the electron spin is nearly perfectly initialized.

The previous experimental scenario would place a UDW detector in a non-chiral Luttinger liquid. To place it in a helical Luttinger liquid (HLL), we could consider HgTe/CdTe quantum wells in their quantum spin Hall effect phase [67,68] and implant quantum dots acting as qubits close to the edge states. We consider this case in detail in our simulations below (see Figs. 6 and 7). Our edge-state simulations predict a velocity of $v = 0.54 \times 10^6\,\mathrm{nm\,ns^{-1}}$, slower than graphene by a factor of 2. This scenario would thus require a switching time of about 55 fs, somewhat less demanding than the first scenario.



The qubits in the HgTe-CdTe scenario could be either chemically synthesized Qdots embedded in HgTe during deposition, doped silicon Qdots analogously embedded in the HgTe, or Qdots defined by gate electrodes. One could then employ gate-induced initialization and tunneling readout of the electron spins in the Qdots [69, 70], as discussed above.

Working with gate-defined Qdots in HgTe is convenient, but short electron relaxation times are a concern. Silicon Qdots will require more work to embed into the HgTe quantum well, but shallow dopants in silicon are known to be excellent qubits [69, 70]; $T_1$ decreases with field [71–73], but is still favorably long, $T_1 \approx 50$ ms, at $T \approx 100$ mK at $B_0 = 5$ T in natural abundance silicon [73].

To realize UDW detectors in a chiral Luttinger liquid we suggest a third scenaio — a transition metal dichalcogenide (TMD) sample consisting of AB-stacked MoTe$_2$/WSe$_2$ heterobilayers in which a quantum anomalous Hall(QAH) effect was discovered recently [25]. TMDs are potentially excellent materials for quantum information applications owing to the naturally occurring low density of nuclear spins. One candidate for qubits are the antisite defects proposed in Ref. 74. These defects can occur in WSe$_2$, suggesting the experimental scenario in Fig. 5.

We can estimate the velocity of the edge electrons in the QAH phase from the bandwidth $W$ and Moiré lattice constant $a_M$. These are expected to take the values $W \sim 1$ to $100$ meV and $a_M \sim 5$ to $10$ nm [75]. Assuming these are correlated, we can take $W = 1$ meV and $a_M = 10$ nm to get

$$v \sim \frac{W}{\hbar \pi / a_M} = 5000 \, \text{nm} \, \text{ns}^{-1}. \tag{29}$$

The smearing length $\lambda_s$ will have to be at least $a_M$ to make use of the nearly flat bands of the Moiré system. Taking it to be 10 nm, we get a switching time of $t_{sw} = 2$ ps. Hence, Moiré-pattern materials have significantly slower electrons and longer switching time. This time scale places it in the vicinity of picosecond electrical pulses such as those achieved in nanoplasmas [76], junction-mixing scanning-tunneling microscope experiments [55–59], and optically driven photoswitches [59–62].

In the above scenarios a conservative estimate was obtained for the time scales needed to operate the gate. These time scales ranged from 50 fs to 2 ps. In each case, significantly longer

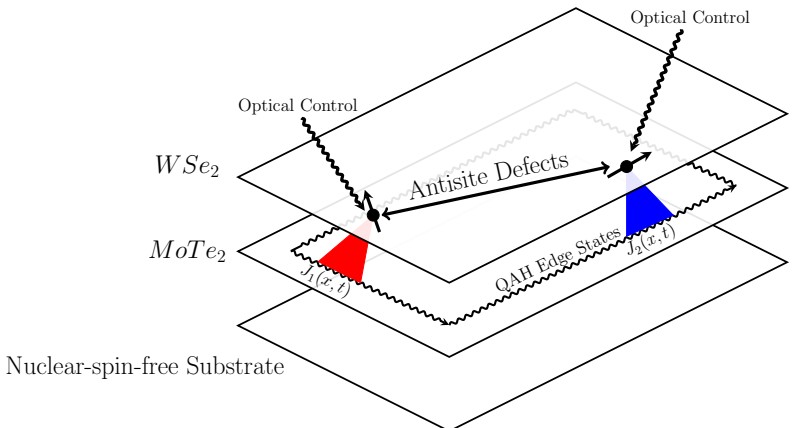

Figure 5: TMD scenario for realizing UDW detectors. Here we envision qubits formed from antisite defects in WSe$_2$ as proposed in Ref. 74 and the quantum anomalous Hall effect edge states as discovered in WSe$_2$/MoTe$_2$ heterostructures. This heterostructure is then placed on a substrate which is ideally not hBN due to the presence of nuclear moments in this material. The UDW detector scenario is then to couple the qubits to these edge states with a controllable couplings $J_1(x, t)$ and $J_2(x, t)$.

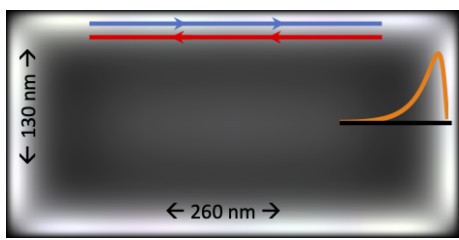

Figure 6: Simulation of HgTe QW edge state on a $200 \times 400$ atom ($130 \times 260$ nm) bar. What is shown is the density of states summed over an energy window of $-10$meV to 10 meV.

time scales would still enable the entanglement of qubits coupled to the field, but the theoretical description of these cases breaks down in this limit. What ultimately limits the experiment is a smearing length of order the coherence length, such as the $2\,\mu$m coherence length of edge states in HgTe QWs [77]. At this length scale, the switching time-scale estimates above would be increase by several orders of magnitude, possibly reaching nanoseconds. At this times scale, electrical gating can be straightforwardly implemented using electrical waveguides and voltage pulses.

By conducting experiments inspired by these scenarios, strong evidence could be obtained for the realization of UDW detectors, which could be used to study of quantum information flow in condensed matter systems and potentially implement all-to-all connectivity in a solid-state quantum computer. Furthermore, the proposed designs could be operated, with considerably less stringent switching requirements, as a long-range all-to-all connected qubit network. This network would, by itself, already be a singular technical advancement.

## 5 Simulating the gated edge state

In Sec. 3, we established gates describing Unruh-DeWitt detectors in Luttinger liquids and identified those that allow for non-zero quantum information channels. Conveniently, the materials that exhibit the phenomenon associated above are achievable and well understood in a laboratory setting. If the velocity of the edge mode is low enough, it will allow for electrical control of the gates, though this will require picosecond electronics such as those using nanoplasmas [76]. We aim to simulate such control in this section with the practical consequences of achieving electrical control an all-to-all connected solid-state quantum information processors or the study of quantum information flow in quantum materials.

Among the three experimental scenarios presented in the previous section, here we will consider the case of a CdTe-HgTe-CdTe quantum well (HgTe QW) in its quantum spin Hall phase whose microscopic parameters are well-known from experiment. It has topologically protected HLL edge states that govern electron transport with an insulating bulk. These states are coherent over [77] 2 $\mu$m, a scale achievable with simulation.

The simulations are carried out using the Bernevig-Hughes-Zhang(BHZ) model [78] placed on a lattice following Ref. 79. It has a mass parameter $M$, a band width controlling parameter $\epsilon$, and a hybridization parameter $\lambda$. The tight-binding Hamiltonian on the square lattice with periodic boundary conditions is

$$H_{\text{TB}}(\mathbf{k}) = \big(M - \epsilon(\cos k_x + \cos k_y)\big)\Gamma_5 + \lambda \sin kx\,\Gamma_x - \lambda \sin ky\,\Gamma_y\,, \tag{30}$$

where, using two sets of Pauli operators $\sigma_{x,y,z}$, $\tau_{x,y,z}$, acting on spin and orbital indices respectively, $\Gamma_5 = I \otimes \tau_z$, $\Gamma_x = \sigma_z \otimes \tau_x$, and $\Gamma_y = -I \otimes \tau_y$. The three parameters map to the parameters in the BHZ continuum model *via* $\epsilon \to -2B$, $M \to -4B + M$, $\lambda \to A$, where the

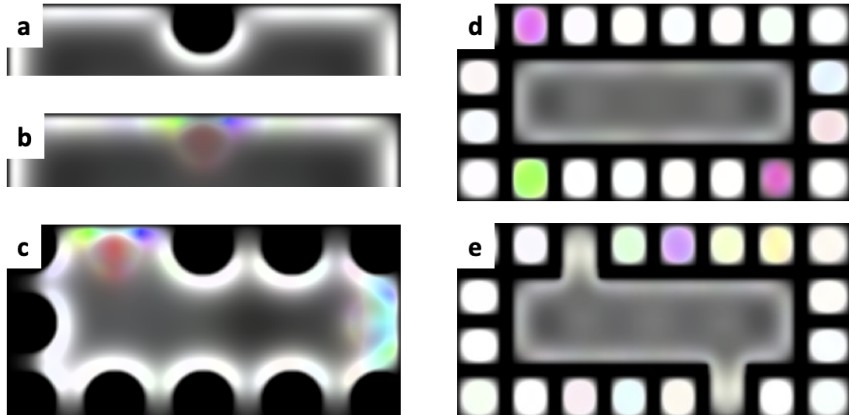

Figure 7: A new electronic quantum bus for spin qubits. **a** A gated region moves the edge state. **b** A local magnetic field penetrates the edge state in two pieces. **c** A snapshot of a 12 qubit all-to-all connected device, 10 qubits at 15.0 V local gate (appear dark in these electron density plots), two qubits at 0.0 V gate exposed to the edge state. **d** A 20 qubit device with gates placed in a regular grid to create trapped electron qubits. **e** Releasing the gate between two of the qubits and center region enables communication.

$M$ in the Lattice model of Ref. 79 is a different parameter than the $M$ in the continuum BHZ model. We fix these parameters to those of the sample described by the 9th row of Table 1 in Ref. 80: $M = -14.6$ meV, $A = 0.55$ eV, $B = -1.87$ eV so that $\epsilon = 3.74$, $M = 7.4654$, and $\lambda = 0.55$. Finally, we switch to open boundary conditions, add a gate potential and local magnetic fields to simulate the presence of a qubit, and generate sparse Hamiltonian matrices of size $320,000 \times 320,000$ whose low energy eigenstates can be obtained using the Arnoldi method on a 200 x 400 lattice of real dimensions 130nm × 260 nm.

An example of these edge states in a simulation is shown in Fig. 6. This simulation reveals the density profile of the edge state for states between -10 meV and 10 meV. It does so by adding up the magnitude squared of the eigenstate wave functions on each site. The results shows edge states with a width of about 20 nm in the inset and illustrates propagation directions at the top with spin up moving to the right and spin down moving to the left.

As shown in Fig. 7(a), if we apply a gate voltage on the edge, we do not destroy the edge state but merely force it to propagate around the gated region. Namely, a gate can be used to programmatically move the "wires" around. In this figure, we used a gate voltage of 15 V to produce the semicircular dark region.

The above simulation could be upgraded to show quantum transduction by placing a spin qubit on the edge, which is represented by a localized magnetic field in the simulation shown in Fig. 7(b), where color denotes the spin direction, we see that the edge state is penetrated at the location of the qubit. This effect is consistent with the "Kondo Insulator" phase induced by a strongly coupled spin impurity on the edge of quantum spin Hall insulators [81]. In HgTe QW and other quantum spin Hall insulators, this penetration of the edge state has likely been observed by the loss of coherence or finite Hall resistance due to spin impurities lying in the vicinity of the edge. The edge states are known to have a coherence length of [77] 2 $\mu$m. But the quantum dot scenario would engineer such behavior and place it under control so long as impurities are very sparse and the coherence length is longer than the edge state, as assumed in this simulation. With a cleanly penetrated edge state, quantum information carried by electrons will not pass by the qubit and instead terminate in or emanate from the qubit introducing a strong coupling between the qubit and edge state.

We now use the simulation to demonstrate all-to-all connectivity. In Fig. 7(c), we placed 12 cavities around the edge of the sample, each with a local gate controlling their coupling to the edge state, each capable of housing a Kondo-like impurity. We selectively enable edge state propagation to/from two of the cavities. Due to the quantum information transmissibility of this channel, turning this coupling on for a short switching time, as discussed in the previous sections, enables the transfer of quantum information between the hypothetical qubits without allowing this information to spread far beyond the qubit during the application of the gate. Hence, in principle, this approach further promises applications of high-fidelity gate operations on just these two cavities, for the edge state circumnavigates all of the other cavities.

In practice, there is an engineering challenge to making qubits that are good at penetrating the edge state to enable high-fidelity gates. One option is to study different spin impurities placed on the edge states and study them using scanning tunneling spectroscopy. Another is to work with Qdots and exploit the many years of research that have gone into engineering their properties. This approach requires designing a suitable dot that can penetrate the edge state (a possibility due to the long time an electron spends in the dot [82]).

An alternative device is shown in Fig. 7(d). Here gates at 15.0 V create the dark regions and trap the electrons into 20 Qdots surrounding the outside of the system. Namely, this system replaces the proposed Si Qdots of 7a-c with trapped electrons. If now a gate is altered near two of these dots (Fig. 7(e)), they connect with the grey region in the middle which holds conducting edge states that propagate on the boundary of this interior region. Similar to 7(c), this connection translates into a gate operation between just the two qubits as it is turned on and off. But now, the concern of whether the dot penetrates the edge state is replaced with the degree to which we can control the tunneling of electrons in and out of the Qdots.

# 6  Future investigations

We focused the experimental proposal on HgTe quantum wells because these are well-known and studied. But they are also hard to synthesize. Alternative materials include GaAs quantum wells in large magnetic fields exhibiting the quantum Hall effect [83], Moiré pattern materials exhibiting either the quantum anomalous Hall effect or quantum spin Hall effect [25], and even ultra-clean single-wall nanotubes suitable for quantum information applications.

Beyond the engineering opportunities, we find a large list of theoretical gateways opened through this exploration. Some of these include; calculating channel capacities of novel quantum information channels, investigating and simulating the balance of the coherence lengths to our gaussian-like time scales of the switching function, and understanding the zero modes of our bosonization formalism and how that may play a role in quantum information propagation. Furthermore, theoretical explorations into systems exploiting atomic quantum dots implanted into Bose-Einstein condensates have been carried out in Refs. 84 and 85. The atomic quantum dots behave as UDWs, similar to our own, and exhibit decoherence that compares analogously to the decoherence of quantum information in an expanding cosmological model of the universe. Extensions to our work could provide an information theoretic approach to understand this problem deeper.

# 7  Conclusion

In this letter, we aimed to propose an experimental approach coupled with a theoretical understanding of a novel quantum computer. The Unruh-DeWitt detector model was deployed as a means to explain quantum information metrics for the interaction between our Qdots and

helical Luttinger liquid. This unification provided a library of unitary gates that allow us to process quantum information through our system. To understand the potential of these gates we evaluated the simplest "toy-model" like gates and demonstrated they produce well known results of high-capacity quantum channels. We showed variations in the theory that would provide channels for processing quantum information in gates that more realistic physically and that are not inherently entanglement-breaking.

Furthermore, we showed that the helical Luttinger liquid HeTe, with a controlled delta-like interaction of a Qdot CdTe, gives us an experimental vision of how these Dirac fermions can propagate as flying qubits. Further investigations are being carried out to not only bridge the gap between these previously disconnected fields of physics but to understand how connecting them in the methods presented in this paper can lead to new and exciting physics.

# Acknowledgments

We thank Charles Kane, Justin Kulp, Jiye Fang, Pegor Aynajian, Yuan Ping, David Klotzkin, Wei-Cheng Lee.

**Funding information** This material is based upon work supported by the National Science Foundation under Grant OAC-1940260.

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
