# Peer review of "Design Constraints for Unruh-DeWitt Quantum Computers"

_SciPost Physics, doi:SciPost Phys. Core 7, 019 (2024)_

## Round 1 · Referee Report · Anonymous (Referee 2) · 2024-2-19

Strengths

see original report and below.

Weaknesses

see original report and below.

Report

I thank the authors for their detailed reply to my comments and the corresponding changes they made to the manuscript.

- I appreciate the toning down of the introduction, the added discussion on spin-boson coupling and Luttinger Liquid physics, and the consideration of complications in implementing gates.

- I regret that the authors did not take this opportuntity to add a more realistic discussion and simulation of the proposed protocol and the experimental constraints it implies, and instead insist these points "will be obvious to an experimentalist interested in quantum computation". Besides it probably being optimistic to think that any interested reader will be familiar with all techniques mentioned in the manuscript, it also denies the general (non-expert) reader the chance to appreciate the actual reach and limitations of the current proposal. As becomes clear from the author's reply letter, any realistic implementation of their proposal will be challenging in many ways.

Although the authors did not include a more realistic discussion of the experimental requirements in their manuscript, and although they chose to include only the simulation of a standard result and not their own protocol, a fuller discussion of these points will be available to interested readers in the (openly available) SciPost review record, which will be published alongside the article. I therefore don't think this should impede publication.
Because of the generic nature of the model and the lack of a realistic discussion and simulation, however, I recommend publication in SciPost Physics Core rather than SciPost Physics.

Requested changes

N/A

---

## Round 1 · Author Response

We thank the referee for this thoughtful and thorough report. We have revised the manuscript to explain better some of the positions he/she/they have presented. Below, we explicitly address the referee's points and provide a sense of how the manuscript was changed. Please also see the latexdiff for details on all the changes.
I will also give some context here to a few of the points you have mentioned so that we are all on the same page and hopefully, we can show you that the physics was correct, but our explanations may have been lacking in these areas.

"The central idea is coupling a qubit to a (helical) 1D bosonic field acting as a quantum channel. This is done by a Hamiltonian such as the one in Eq 2 or 5. These Hamiltonians are completely standard descriptions of spin-boson coupling. It is unclear why the authors insist on calling them "Unruh-DeWitt detectors" when they are not used as detectors in any way; when there is no Unruh physics involved; and when there are many other, more appropriate (standard) ways of describing the same types of spin-boson coupling."

One of the defining features of this paper is that the Hamiltonian in Eqs. 2 and 5 is familiar to condensed matter physics but also of the same form as the Unruh--DeWitt detector Hamiltonian. However, our Hamiltonian has some subtle notational differences. I have adjusted the manuscript to emphasize this relation.

In some contexts, UDW detectors are utilized to couple qubits to quantum fields without the context of the Unruh effect (See. Simidzija et al.). Though an interesting study would be to elucidate a correlation of the Unruh effect and the simpler context presented in this manuscript.

"The proposal to use the bosonic degrees of freedom in Luttinger Liquids to couple qubits to, is a good idea. But the degrees of freedom you couple to are then really bosons, and the proposal is not very different from earlier proposals for coupling qubits through bosonic channels. The proposed channel certainly does not use a "strongly coupled fermionic system" to act as a channel, as claimed in the introduction. In fact, it uses precisely the bosonic (LL) channel that is left over after all strongly correlated electron physics at high energies has been integrated out. Moreover, the fact that a bosonic field can yield non-zero channel capacity and even perfect communication is a previously known result, cited by the authors. It is therefore unclear which part of the current proposal really represents a new ingredient in the search for efficient quantum channels."

Thank you for making this point. Bosonization exploits the inability of particles in one dimension to braid and so exhibit their statistics. It establishes the equivalence between fermionic and bosonic operators at the level of correlators. As it turns out, the quantum channel of interest traces over the field in such a way that the quantum channel and quantum capacities depend only on these field correlators. Therefore, when evaluating the quantum capacity of the channel, we can ensure a direct equivalence between fermionic and bosonic operators and so begin with a Kondo coupling, but end with qubits communicating via bosonic modes.

We agree we oversold this point in the first version of the manuscript. We understand that there is subtle physics, specifically in the regime of Kondo-like impurities coupled to LLs that could inhibit the methods used to bosonize to the Hamiltonians that we did. So, the result is not surprising given the choice of Hamiltonian, and the literature on the subject. However, it is of immense experimental importance for it implies an electron gas of some 2D materials has the potential to transport quantum information perfectly up to the subtle corrections just mentioned. We have adjusted the manuscript to emphasize these points.

"Although the authors define gate operations in section 3C, they omit a discussion of how these would be implemented in any concrete setting. Especially gates of the form of eq 20 do not seem to conserve the total charge in the LL, and seem at first sight hard to implement in any concrete way. Additional discussion on this point would be welcome."

I have adjusted the manuscript to recognize the difficulty in implementation of these gates by elaborating on complications involved in our model. I also indicate possible avenues to overcome these complications for future studies.

"The whole proposal heavily depends on the availability of an actual Luttinger Liquid in real systems. Although the three experimental settings mentioned by the authors are good candidate systems for Luttinger Liquid physics, and a Luttinger Liquid is expected to be present in them at very low temperatures, real unambiguous experimental demonstrations of Luttinger Liquid physics in these (and other) systems are rare. Finite-size corrections to Luttinger Liquid physics are crucial in most practical setups, and are likely to also strongly influence the proposed quantum channels. If the authors could provide a quantitative analysis of how robust their proposal is to such effects, and what the expected influence of finite size effects would be in their proposed setups, this would make their proposal much more concrete."

These finite corrections heavily depend on the nature of the coupling, which is novel to this system given the rapid switching on and off of the coupling between the Kondo-like impurity and the Luttinger liquid. We agree with the reviewer that the finite-sized corrections to the Luttinger liquid could strongly influence the quantum information of our idealized system proposed in our model. Currently, we are developing a formalism to understand a near-term quantum device that will allow for first principle investigations into this coupling. We hope that through these further studies we might gain insights to how this protect against these effects in a system that can both realize our ambitions, but also be stable in a laboratory setting.

Regardless of this, we demonstrate the first design constraints given the idealized situation. We expect there to be far more constraints introduced in the future as the project continues to develop.

"The discussion of how to implement the proposal in three possible experimental settings is a welcome part of the manuscript. However, in this discussion the authors selectively combine several different techniques at the cutting edge of current developments in their respective fields. I expect that using femto-second voltage pulses, which can controllably select and target just one of several nm-sized qubit separated by microns, inside a cryogenic environment, and with the possibility of simultaneously applying a 9T magnetic field, will be technically challenging. This makes the presentation of the experimental implementations less concrete than the estimates of engineering constraints presented by the authors seem to suggest. A more realistic discussion on this point would certainly improve the presentation of the current proposal."

The reviewer is correct that our proposals for achieving long-range coupling between solid-state qubits are experimentally demanding. We have been careful to enumerate what we believe are the main technical challenges in each approach, and we have cited appropriate prior work to indicate feasibility. To those "skilled in the art," we feel enough detail is provided to lay out an experimental plan.

All the proposals require high field and low temperature to achieve near unity electron-spin spin polarization. Achieving fields up to 9 T is routine using superconducting magnets. A sample temperature of 4.2 K is easily achieved in a liquid-helium-immersion cryostat, and 2.1 K is achieved by pumping on the associated helium bath. A temperature as low as 20 mK (at fields of 9 T) is achieved using expensive closed-cycle dilution refrigerators. In summary, the requisite temperatures and fields are feasible.

The real challenge is delivering microwaves, to a cryogenic apparatus, that are resonant with electron spin flips at 9 T. For a g = 2 electron, the associated resonance frequency is 252 GHz. Delivering a microwave magnetic field strong enough to provide sub-nanosecond spin flips at this frequency using traditional centimeter-scale metallic waveguides is feasible at 2.1 K, but becomes infeasible at 20 mK due the limited cooling-power budget of cryogenic coolers at millikelvin temperatures. In the manuscript we therefore discuss using chip-scale microwave sources. These chip-scale sources are submillimeter in size and are therefore more easily heat-sunk. Moreover, they can be coupled to a coplanar waveguide, tapered to a submicron width, that should generate a sufficiently high magnetic field within the limited power budgets available at 2.1 kelvin, certainly, and possibly even at millikelvin temperatures. We cite prior work on chip-scale microwave sources operating above 252 GHz and at room temperature, and operating at 20 GHz and 1 K. A chip-scale microwave source operating at both 252 GHz and 2.1 K has not yet been demonstrated, but is technically feasible. Creating high-frequency chip-scale microwave sources operating at 2.1 K is a topic of active research. See, for example, co-author Marohn's NIH grant with electrical engineer Ehsan Afshari at the University of Michigan, Grant 5R01GM143556.

All the proposals require nanometer-scale electrical gates. The width of the graphene nanoribbon in Fig. 4, for example, is 30 nm. The required gate pitch would therefore also be 30 nm. Each gate would ideally double as the center line in a coplanar waveguide used to deliver resonant microwaves to the qubit. A 30 nm gate pitch could be achieved with a coplanar waveguide having a 5 nm centerline, a 5 nm spacing, a 15 nm ground plane, and another 5 nm spacing. The critical dimension here is 5 nm, which is larger than 2 nm resolution limit of electron-beam lithography circa 2013. So creating the proposed gates, embedded in coplanar waveguides, is likewise feasible, although expensive and time-consuming. The specifications in the above paragraph and the feasibility of doing few-nanometer-resolution electron-beam lithography would be obvious to an experimentalist interested in quantum computing, and so we decided not to include them in the manuscript.

The reviewer should also appreciate that each of the proposed systems is chemically challenging. Graphene nanoribbons 100's of nanometer in length can be synthesized but quickly become insoluble as their length increases; this is one of the reasons that the Ref. 11 proposal has not been implemented. We note there was recent DoD MURI grant call on graphene-nanoribbon quantum computing. Making the proposed HgTe quantum well is likewise non-trivial, requiring an experienced material grower operating a dedicated mult-million-dollar molecular-beam epitaxy machine. Because Te is persistent in the chamber and will poison most other inorganic semiconductor devices, there are only a few such dedicated machines available worldwide. How to embed the proposed quantum dots in the HgTe quantum well, at well-spaced distances, is non-obvious. This is a challenge that would require a large team and major grant to solve. Stacked semiconducting monolayers, like the WSe2 and MoTe2 layers shown in Fig. 5, have now been achieved by scores of groups. However, creating and finding anti-site defects in these materials is a major experimental effort by itself and, again, a topic of active research.

We feel that enumerating ideas for practically producing the proposed graphene nanoribbons, HgTe quantum wells, and dichalcogenide structures is beyond the scope of the present manuscript.

"The simulation presented in section V is not actually a simulation of the proposed quantum gates, nor of the proposed communication through the quantum channel, nor even of a Luttinger liquid. The simulation only shows topological edge states present in a prototype model for a topological insulator, under various boundary conditions. The outcomes are not unexpected and reproduce well-known results from the literature. It is thus unclear what these simulations add to the proposal. I would recommend that the authors simulate the quantum gates and quantum communication protocols they propose. Showing that numerically they can achieve efficient coupling between qubits and channel, lossless communication for quantum information between qubits, and programmable all-to-all connectivity would strengthen the manuscript."

You are correct that the simulation in Figure V does not include quantum information transduction between qubits and fields, the main emphasis of the paper. However, this simulation does show the ability to turn on and off the interaction in HgTe. Notably, it shows the ability to bend the edge states around a cavity and demonstrates the design constraints necessary to turn on and off the interaction with the cavity. This demonstration is non-trivial because the Klein paradox predicts that Dirac fermions typically penetrate potential barriers unimpeded. Since all parameters were realistic, the voltage used for the gates shows that the simulations predict experimental control over the Kondo coupling between edge states and stationary qubits. Nevertheless, we agree these results are unsurprising even though they are new (to the best of our knowledge) and highly relevant to our work. I have adjusted the manuscript to note these points and have discussed that including the impurities as spin-qubits would align with the well-known literature you mentioned.

Again, we appreciate the comments. Thank you for your time.

Yours Sincerely,

Eric W. Aspling, John A. Marohn, and Michael J. Lawler

---

## Round 1 · List of Changes

For a pdf of the article with all of the highlighted changes please see: https://drive.google.com/file/d/1c121rKPPPb2zeWT_XNYbSFOqi8C3_OV7/view?usp=drive_link

---

## Editorial Decision

published